# Effectiveness of Mediterranean Diet Implementation in Dry Eye Parameters: A Study of PREDIMED-PLUS Trial

**DOI:** 10.3390/nu12051289

**Published:** 2020-05-01

**Authors:** Ignacio Molina-Leyva, Alejandro Molina-Leyva, Blanca Riquelme-Gallego, Naomi Cano-Ibáñez, Laura García-Molina, Aurora Bueno-Cavanillas

**Affiliations:** 1Department of Ophthalmology, Hospital Virgen de las Nieves, 18014 Granada, Spain; igmoley@gmail.com; 2Department of Dermatology, Hospital Virgen de las Nieves, 18014 Granada, Spain; 3Department of Preventive Medicine and Public Health, Universidad de Granada, 18016 Granada, Spain; blanca.riquel@gmail.com (B.R.-G.); ncaiba@ugr.es (N.C.-I.); lagarmol1@gmail.com (L.G.-M.); abueno@ugr.es (A.B.-C.); 4Instituto de Investigación Biosanitaria de Granada IBS, 18012 Granada, Spain

**Keywords:** mediterranean diet, dry eye, fatty acids, extra virgin olive oil, PREDIMED-PLUS trial

## Abstract

The purpose of this study is to evaluate the effect of a Mediterranean diet supplemented with extra virgin olive oil and nuts on dry eye parameters. The participants in this study were randomized into one of the two interventional arms: (1) a standard intervention group, a Mediterranean diet supplemented with extra virgin olive oil and nuts; and (2) an intensive intervention group, based on a hypocaloric Mediterranean diet and an intensive lifestyle program with physical activity and weight-loss goals. In both groups, common dry eye tests were conducted at baseline and after six months: the Ocular Surface Disease Index (OSDI), the Dry Eye Scoring System (DESS), tear break-up time (TBUT), the Schirmer’s test, and the Oxford staining grade. Sixty-seven eyes were examined. After six months, dry eye parameters improved in both groups; differences between groups were favorable for the intensive intervention group. The implementation of a Mediterranean diet pattern was beneficial for the selected patients with dry eye, and could be beneficial for patients with dry eye in general. Behavioral support for diet adherence and the promotion of healthy lifestyles (exercise) and weight loss (calorie restriction) have an added positive effect.

## 1. Introduction

The management of dry eye syndrome (DES) can be a major concern for ophthalmologists, owing to its high prevalence and the difficulty of prescribing an appropriate and acceptable treatment for the patients. It represents 25% to 80% of all ophthalmological conditions and is usually an additional complaint that many patients refer to aside from their main ocular disease, producing an important impairment of quality of life [1,2].

For many years, the treatment of dry eye has been based on two pillars: (1) replacing the deficit using artificial tears, and (2) correcting any identifiable causal factors. However, many patients are reluctant to apply artificial tears, and usually, a causal factor is not easy to identify [3]. Another option for treatment is oral supplements containing free fatty acids that have shown the potential to improve symptoms and increase the quality of life of subjects with dry eye [4]. However, their use is limited by their price or the inconvenience of taking pills several times a day.

New scientific insights on the disease have started to bring to light its pathogenesis. On the one hand, tear secretion and the ocular surface can be altered by different factors. It is probable that in most cases, etiology is multifactorial, leading to different subtypes of dry eye, depending on the predominant aggressor [5]. Aging and oxidative stress are likely to be two of the most important factors due to their frequency and relevance. Reactive oxygen species produced for external injuries may trigger dry eye pathology that lead to chronic inflammation and cell damage [6]. Among the conditions related to dry eye, we found metabolic syndrome [7,8]. Metabolic syndrome is associated with a systemic pro-inflammatory status and causes lacrimal hyper-osmolality that is believed to activate inflammatory pathways and initiate cytokine release, causing corneal damage and loss of goblet cells [9].

Few studies have analyzed the correlation between healthy lifestyle interventions and dry eye disease. These studies showed an improvement in dry eye patients associated with weight loss, calorie restriction and active lifestyle [10,11,12]. The Mediterranean diet (MedDiet) pattern contains naturally occurring fatty acids and other elements that we found in the oral supplements. In addition, it has shown strong evidence in terms of cardiovascular risk reduction, the promotion of a systemic anti-inflammatory status, weight loss, and the reduction in metabolic syndrome prevalence [13,14,15]. Therefore, we posed the question: could the adherence to a MedDiet pattern improve dry eye syndrome?

The aim of this study was to analyze the effect of the Mediterranean diet pattern on dry eye syndrome in patients with metabolic syndrome. The secondary aim was to evaluate the effects of two types of dietary intervention—standard vs. intensive.

## 2. Materials and Methods

### 2.1. Study Design

This study was conducted in the frame of the “PREvención con DIeta MEDiterránea” (PREDIMED-PLUS) trial. PREDIMED-PLUS is a multicenter, randomized, primary prevention field trial of cardiovascular disease in community-dwelling men (aged 55 to 75 years) and women (aged 60 to 75 years) diagnosed with metabolic syndrome, with a body mass index (BMI) ≥ 27 and < 40 kg/m^2^. The study protocol was conducted according to the guidelines laid down in the Declaration of Helsinki, and all procedures were approved by the steering committee of PREDIMED-PLUS. Written informed consent was obtained from all the study participants. A total of six centers from Granada participated in the study. Patients were randomized in two groups—standard vs. intensive intervention.

### 2.2. Inclusion and Exclusion Criteria

The inclusion criteria were: (1) PREDIMED-PLUS study enrolment, (2) a positive history of ocular surface discomfort, and (3) the provision of informed consent. The exclusion criteria were: (1) an active allergy or infection at the ocular surface, (2) patients currently using an ophthalmic medication, except for artificial tear preparation or eye cleaning solution, (3) the use of topical steroidal or non-steroidal anti-inflammatory drugs, (4) ocular surgery in the last 6 months, (5) the consumption of systemic medication that may interfere with tear production, including anti-depressive, anti-anxiety, or anti-histaminic drugs, and (6) an allergy to fluorescein or topical anesthesia.

### 2.3. Study Intervention

The study intervention was the use of a Mediterranean diet supplemented with extra virgin olive oil and nuts. The standard group included no other interventions. The intensive group included an additional 30% calorie restriction, an intensive lifestyle program with physical activity (walking 45 min/day or equivalent), and weight-loss goals with behavioral support. More information about the PREDIMED-PLUS trial is available at http://predimedplus.com.

### 2.4. Main Outcomes of Interest

The study consisted of two ophthalmological assessments performed at baseline and at 6 months. Diet adherence, adverse effects, and undercurrent diseases were collected at all study visits. The ophthalmological assessment consisted of two questionnaires: Dry Eye Scoring System (DESS), a six-item test that measures the frequency of dry eye symptoms (burning sensation, grittiness, redness, blurred vision, ocular fatigue, and excessive blinking) and Ocular Surface Disease Index (OSDI) a twelve-item test that measures the frequency of dry eye symptoms in daily life activities. And an external eye examination performed by an experienced ophthalmologist. The external eye examination included: visual acuity, a slit-lamp examination searching for signs of meibomian gland dysfunction, staining score, tear break-up time (TBUT), and Schirmer’s test type 2. All examinations were performed under the same environmental conditions.

Staining scores were assessed using 10 microliters of fluorescein sodium (Alcon Laboratories Inc., TX, USA) measured with a micropipette and instilled in the superior conjunctiva. After a few blinks, patients were examined with a slit-lamp using a cobalt blue light and a yellow filter that enhances fluorescence. Both eyes were examined and classified according to the Oxford staining score.

TBUT was evaluated after assessing the staining score, measuring the time between the last blink and the appearance of black gaps in the tear film. Three measures were done, and the average result was calculated. Both eyes were examined.

Schirmer’s test was performed after a washout period of ten minutes. Fifteen microliters of oxybuprocaine 1 mg/mL + tetracaine 4 mg/mL were instilled (Alcon Laboratories Inc., TX, USA). After a minute, the tear excess was removed, and a milimetered strip was placed in the conjunctival sac. The results were measured after five minutes in both eyes.

Diet parameters were collected through physical examination, a food frequency questionnaire, and blood samples. The main variables were body mass index, abdominal perimeter, triglycerides, total lipids, HDL cholesterol, total cholesterol, saturated fatty acids, monounsaturated fatty acids, polyunsaturated fatty acids, n-3 omega fatty acids, and n-6 fatty acids.

### 2.5. Statistical Analysis

Descriptive statistics were used to explore the characteristics of the sample. Continuous data are expressed as mean ± standard deviation (SD). The absolute and relative frequency distributions are estimated for qualitative variables. The Wilcoxon–Mann–Whitney test was used for comparisons between the type of intervention and the independent continuous data. The Wilcoxon signed-rank test was used for comparisons between the type of intervention and the paired continuous data. The χ^2^ test, or Fisher’s exact test, were applied when necessary for comparisons between the type of intervention and the nominal variables. The significance was established for all two-tailed tests, *p* < 0.05. Statistical analyzes were performed using SPSS version 22, (SPSS, Chicago, IL, USA).

## 3. Results

A total of fifty subjects from the PREDIMED-PLUS group were contacted and offered to participate in the current study. Thirty-four subjects agreed to join this study. All the participants completed the study protocol without any adverse reactions. Thirty-four subjects, who represent 67 eyes, were included for the analyses. Forty-two eyes corresponded to the intensive intervention group, and 25 to the standard intervention group.

### 3.1. Anthropometric and Biochemical Variables

We analyzed anthropometric characteristics as well as biochemical variables related to metabolic syndrome and fatty acid consumption. The mean age of the study subjects was 64.47 ± 5.19 years, range 55–74, with a comparable age distribution between the standard and intensive intervention groups (*p* = 0.23). The proportion of women was 61.8% (*p* = 0.14). The BMI and the abdominal perimeter experienced a significant reduction after six months of intensive intervention (*p* < 0.05). The main characteristics of the study participants are summarized in Table 1.

After six months of intervention, of all the biochemical parameters studied, glucose and total cholesterol showed no significant differences. We found a reduction in blood triglyceride levels: 17.62 ± 40.42 mg/dl (*p* = 0.02), and an increase in HDL cholesterol levels: 2.37 ± 5.03 (*p* = 0.02), but no significant differences were observed between the study groups (Table 2).

The analysis between the groups of the fatty acids ingested in the diet showed a significant increase in the intake of mono- and polyunsaturated fatty acids in the intensive intervention group, and a slight reduction in the standard group. When we analyzed in detail the main fatty acids, we observed a reduction in the consumption of omega 6 and omega 3 fatty acids in the standard intervention group, without it being a significant difference.

### 3.2. Dry Eye Parameters

Symptoms, TBUTs, ocular surface staining, and the Schirmer’s tests recorded in the first study visit after randomization are shown in Table 3. No significant differences were found between the groups at the beginning of the study.

Regarding the subjective tests that analyzed the appearance of symptoms related to dry eye, the Mediterranean diet intervention showed an improvement in dry eye symptoms with reductions in the DESS test score, −0.35 ± 0.15 (*p* = 0.025), and the OSDI, −1.75 ± 0.9 (*p* = 0.039). However, there were no significant differences between the types of intensive versus standard interventions (Table 4).

The tear stability increased by a mean of 0.67 ± 0.15 s (*p* = 0.01) after the study period. The intensive intervention group showed a greater improvement in lacrimal stability compared to the standard intervention: 0.95 ± 1.3 versus 0.23 ± 0.9 s (*p* = 0.02). The mean increase in the Schirmer’s test with anesthesia from the beginning was 0.5 ± 0.14 mm (*p* = 0.01). The intensive intervention group experienced better performance in the Schirmer’s test compared to the standard intervention: 0.72 ± 1.2 versus 0.12 ± 0.8 mm (*p* = 0.03). The ocular surface staining, applying the Oxford grading scale, showed a decrease of 0.10 ± 0.04 points (*p* = 0.01). The reduction was greater in the intensive intervention group compared to the standard group: 0.16 (SD 0.37) and 0 (SD 0), respectively, *p* = 0.03.

## 4. Discussion

Our study evaluates the effectiveness of the implementation of the MedDiet on dry eye in subjects with metabolic syndrome. Additionally, it compares two types of implementation, standard vs. intensive, which adds behavioral support, calorie restriction, and healthy lifestyle recommendations. The MedDiet pattern implementation showed an overall improvement of dry eye signs and symptoms. The intensive intervention showed a better improvement of dry eye signs compared to standard intervention.

The MedDiet is characterized by a high intake of fruits, vegetables, legumes, nuts, cereals, fish, and olive oil (coupled with low intake of saturated fats); low intake of meat and dairy products; and regular but moderate intake of alcohol, mostly wine [16]. The beneficial effects of olive oil have been attributed to its high content of oleic acid, a type of monounsaturated fatty acid (MUFA), as it protects against oxidative damage [17]. In addition to MUFAs, phenolic compounds also exhibit antioxidant and anti-inflammatory effects, commonly associated with the origin of major chronic diseases. As one of the major phenolic compounds present in virgin olive oils, hydroxytyrosol presents a variety of pharmacological activities, such as antioxidant properties and anti-cancer, anti-inflammatory, and neuroprotective activities, as well as beneficial effects on the cardiovascular system [15,18].

Essential fatty acids (EFAs) are necessary to maintain good health and cannot be synthesized by human beings. The most important polyunsaturated fatty acids are omega-3 (n-3 PUFA) and omega-6 (n-6 PUFA), which play an important role in regulating the inflammatory and immune responses. Humans evolved on a diet in which the n-6:n-3 ratio was approximately 1:1. However, the current Western diet tends to be deficient in n-3 PUFA, and this ratio is typically much higher (approaching 17:1) [19]. A systemic low-level of omega fatty acids is a risk factor for dry eye disease [20]. The metabolism of EFAs generates molecules with anti-inflammatory properties [21]. Studies using nutritional supplements containing omega-3 and omega-6 fatty acids have shown to be beneficial for patients suffering with dry eye [4]. Ziemanskin et al. conducted a similar study, analyzing the relationship between dietary essential fatty acid intake, dry eye disease, and meibomian gland dysfunction in a group of post-menopausal women. They observed that the dietary consumption of n-3s and n-6s showed no association with dry eye disease, but that high n-3 consumption and moderate n-6 consumption were protective against meibomian gland dysfuntion [22].

It is probable that the underlying mechanism between the MedDiet and the improvement of dry eye is based on a reduction in the inflammatory process in the ocular surface. Pre-clinical studies have revealed the effect of PUFAs on cellular models. The treatment of humanized meibomian gland epithelial cells with n-3 fatty acids produced an up-regulation of lipid [23]. A combination of n-3 and n-6 fatty acids was also shown to increase meibum [24]. These results suggest that n-3 and n-6 fatty acids work directly on the meibomian glands to increase lipid production. Studies on animals have also shown evidence of the influence of PUFAs on the visual system. Retina and lacrimal glands incorporated dietary fatty acids, indicating that both tissues are very sensitive to dietary changes and fatty acids composition [25]. Additionally, lacrimal measurements in rats, after a period of supplementation, showed an increase in tear production and lower ocular surface inflammation [26]. Dietary PUFAs have been shown to be a protective factor against dry eye signs in experimental rats suffering with pilocarpine-induced dry eye. Dietary n-3 PUFA could modify the anti-inflammatory/pro-inflammatory ratio and decrease pro-inflammatory prostaglandin levels in the lacrimal gland. Supplemented animals showed a decrease in clinical signs of chronic corneal dryness [27,28].

In our study, the improvement of dry eye is marked when the MedDiet pattern is applied in an intensive manner, which could have different implications. First, subjects of the intensive group had a coach that encouraged them to adhere to the MedDiet with weekly interviews to assess the achievement of goals: MedDiet pattern, calorie restriction, and exercise. This group of subjects is more likely to show better adherence to the MedDiet pattern. Second, calorie restriction and exercise will probably have an additional effect, particularly in our study population. The patients of our study suffered from metabolic syndrome, a condition associated with a systemic pro-inflammatory status, which, in itself, can contribute to the pathogenesis of dry eye [7,29]. Metabolic syndrome is a common condition in Western countries, affecting about 25% of adults [30]. Weight loss and exercise are meant to improve metabolic syndrome, subsequently reducing systemic inflammation and ocular surface inflammation. This model has been demonstrated in other systemic inflammatory diseases with terminal organ manifestation, such as psoriasis [31]. It is probable that overweight and obese subjects who do not fulfill the criteria of metabolic syndrome will also experience benefits, as abdominal fat itself has a pro-inflammatory effect [28]. More than 30–70% and 10–30% of the population in Western countries are overweight and obese, respectively [32].

We acknowledge some methodological limitations in our study: (1) the lack of a placebo group—however, this could be an ethical problem, as the MedDiet has been proven to have beneficial effects on cardiovascular risk; (2) the sample size is limited; (3) all the patients have metabolic syndrome, therefore the extrapolation of the results to the general population should be made with caution; and 4) there is not a universally accepted gold standard test for dry eye evaluation.

## 5. Conclusions

The results of our study show that the implementation of a MedDiet pattern for 6 months is beneficial for patients with metabolic syndrome suffering from dry eye, and could be beneficial for patients with dry eye in general. Behavioral support for diet adherence and the promotion of healthy lifestyles (exercise) and weight loss (calorie restriction) have an added positive effect. The recommendation of the MedDiet in patients with dry eye is not only harmless but has proven beneficial systemic and cardiovascular effects. The findings of this study are valuable not only for ophthalmologists but also for general practitioners who can give behavioral support and bring additional value to their dietary and healthy lifestyle recommendations: “not only will your cardiovascular health improve, your eye discomfort will too.” More studies are needed, including for the general population, regarding long-term effects and the exploration of other dietary patterns.

## Figures and Tables

**Table 1 nutrients-12-01289-t001:** Baseline anthropometrics and diet characteristics.

	Standard Intervention (N = 13)		Intensive Intervention (N = 21)		
	Mean/N	SD/%	Mean/N	SD/%	*p*
Age	63.23	5.29	65.47	5.18	0.23
Women	7	50%	14	71%	0.14
Weight (kg)	87.39	14.51	85.64	12.65	0.89
Height (cm)	164.69	8.25	162.21	10.40	0.61
BMI (kg/m^2^)	32.07	3.66	32.49	3.15	0.47
Abdominal circumference (cm)	110.19	10.56	106.7	11.36	0.76
HTA (No. subjects)	11	84%	18	85%	0.59
DM-2 (No. subjects)	2	15%	3	14%	0.99
Glucose (mg/dL)	105.30	28.76	102.85	29.67	0.84
Total Cholesterol (mg/dL)	194.69	37.95	202.95	28.98	0.82
HDL Cholesterol (mg/dL)	50.61	13.35	51.71	8.99	0.82
Triglycerides (mg/dL)	133.46	35.41	172	80.83	0.19
Dietary total fat (gpd)	109.45	29.49	100.03	21.76	0.34
Dietary saturated fatty acids (gpd)	26.75	8.06	24.49	7.19	0.43
Dietary monounsaturated fatty acids (gpd)	55.35	15.10	52.44	11.59	0.45
Dietary polyunsaturated fatty acids (gpd)	19.46	7.11	15.96	5.42	0.09
Dietary linoleic acid (gpd)	15.36	5.9	12.06	4.42	0.08
Dietary linolenic acid (gpd)	1.82	0.8	1.37	0.7	0.07
Dietary Omega 3 fatty acids (gpd)	1.19	0.65	1.11	0.47	0.93

BMI: body mass index; DM-2: type 2 diabetes mellitus: gpd: grams per day; HTA: hypertension; No: number; SD: standard deviation; %: percentage. Data are expressed as absolute (N) and relative frequencies (%), and mean and standard deviation (SD). The Wilcoxon–Mann–Whitney test was used for comparisons between the type of intervention and the continuous independent data. The χ^2^ test, or Fisher’s exact test, when necessary, was used for comparisons between the type of intervention and the nominal data.

**Table 2 nutrients-12-01289-t002:** Anthropometrics and diet parameters after intervention.

	Total Differences	*p*	Differences among Groups		*p*
	Mean Difference (SD)		Standard Intervention	Intensive Intervention	
BMI	−0.74 (1.56)	0.01 *	−0.59	−1.78	0.03 *
Abdominal circumference (cm)	−3.33 (5.67)	0.01 *	−0.75	−4.98	0.02 *
Glucose (mg/dL)	−6.70 (21.55)	0.09	−9.30	−5	0.57
Total Cholesterol (mg/dL)	7.55 (23.41)	0.09	14.25	3.31	0.19
HDL Cholesterol (mg/dL)	2.37 (5.03)	0.01 *	3.83	1.36	0.17
Triglycerides (mg/dL)	−17.62 (40.42)	0.02 *	−16.92	−17.26	0.98
Dietary total fat (gpd)	−0.48 (27.09)	0.91	−10.67	6.67	0.06
Dietary saturated fatty acids (gpd)	−2.68 (6.53)	0.02 *	−3.66	−1.86	0.43
Dietary monounsaturated fatty acids (gpd)	4.74 (17.37)	0.12	−2.11	9.20	0.05 *
Dietary polyunsaturated fatty acids (gpd)	1.74 (8.29)	0.23	−1.78	3.92	0.04 *
Dietary linoleic acid (gpd)	1.19 (6.63)	0.30	−1.18	2.70	0.09
Dietary linolenic acid (gpd)	0.30 (1.05)	0.11	−0.06	0.43	0.17
Dietary Omega 3 fatty acids (gpd)	−0.20 (0.62)	0.06	−0.45	−0.03	0.04 *

BMI: body mass index; gpd: grams per day; SD: standard deviation; %: percentage; *: statistically significant. Data are expressed as mean and standard deviation (SD). The Wilcoxon signed-rank test was used for comparisons between the type of intervention and the paired continuous data.

**Table 3 nutrients-12-01289-t003:** Baseline clinical characteristics.

	Standard Intervention (N = 25 eyes)		Intensive Intervention (N = 42 eyes)		*p*
	Mean	SD	Mean	SD	
DESS	4.46	4.13	3.52	2.58	0.4
OSDI	21.28	22.18	18.58	20.02	0.7
TBUT (sec)	6.81	1.84	6.74	3.11	0.92
Schirmer’s test (mm)	6.64	3.04	6.09	2.87	0.46
Staining Score	0.52	0.51	0.79	0.75	0.12

DESS: Dry Eye Scoring System; mm: millimeters; N: number; OSDI: Ocular Surface Disease Index; *p*: *p*-value; SD: standard deviation; sec: seconds; TBUT: tear break-up time. Data are expressed as mean and standard deviation (SD). The Wilcoxon–Mann–Whitney test was used for comparisons between the type of intervention and the continuous independent data.

**Table 4 nutrients-12-01289-t004:** Clinical parameters post-intervention.

	Total		*p*	Among Groups		*p*
	Main Difference	SD		Standard Intervention	Intensive Intervention	
DESS	−0.35	0.15	0.02 *	−0.07	−0.56	0.11
OSDI	−1.75	0.9	0.03 *	−2.04	−1.57	0.81
TBUT (sec)	0.67	0.15	0.01 *	0.24	0.95	0.02 *
Schirmer’s test (mm)	0.51	0.14	0.01 *	0.12	0.76	0.03 *
Oxford staining score	−0.1	0.04	0.01 *	0	−0.16	0.03 *

DESS: Dry Eye Scoring System; mm: millimeters; N: number; OSDI: Ocular Surface Disease Index; p: p-value; SD: standard deviation; sec: seconds; *: statistically significant; TBUT: tear break-up time. Data are expressed as mean and standard deviation (SD). The Wilcoxon signed-rank test was used for comparisons between the type of intervention and the paired continuous data.

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
