# Peer review of "Effectiveness of Mediterranean Diet Implementation in Dry Eye Parameters: A Study of PREDIMED-PLUS Trial"

_nutrients, 2020, doi:10.3390/nu12051289_

Round 1
Reviewer 1 Report
Comments for the authors:
Introduction must be revised by incorporating more recent references, research findings on the topic, and doing the comparison in order to increase the readability of the paper.
The title of all the tables must be written on top of the respective tables.
The authors are advised to present the statistical data more clearly.
Present the limitations and disadvantages of the study.
There are many small paragraphs. All small and related paragraphs must be merged into one to make a bigger paragraph.
The English language and grammatical errors must be revised throughout the manuscript.
Author Response
Introduction must be revised by incorporating more recent references, research findings on the topic, and doing the comparison in order to increase the readability of the paper.
We appreciate your suggestions. Introduction section has been revised adding more recent references:
[5] Bron, AJ, et al. TFOS DEWS II pathophysiology report. (Lines 42-45)
[8] Kawashima, M, et al. Decreased tears volume in patients with metabolic syndrome: the Osaka (Lines 47-48)
[10] Kawashima, M, et al. Impact of lifestyle intervention on dry eye disease in office workers: a randomized controlled trial. (Lines 52-54)
[11] Kawashima, M, et al. The Association between Dry Eye Disease and Physical Activity as well as Sedentary Behavior: Results from the Osaka Study. (Lines 52-54)
[12] Kawashima, M., et al. Calorie restriction: A new therapeutic intervention for age-related dry eye disease in rats. (Lines 52-54)
We have added new content and compared our study approach with similar ones. Text has been reorganised to improve readability of this section
The title of all the tables must be written on top of the respective tables.
Change has been made in all tables (lines 179, 215, 224, 245)
The authors are advised to present the statistical data more clearly.
Statistical analysis section has been rewritten in in order to clarify data analysis (lines 137-144). Detailed information about statistical analysis has been added to each data table and footers with abbreviations and statistical analysis have been added (lines 181-185; 216-218; 227-230, 246-249).
Present the limitations and disadvantages of the study.
Limitations and disadvantages have been specified (lines 335-339)
There are many small paragraphs. All small and related paragraphs must be merged into one to make a bigger paragraph.
We have taken your concern in consideration. We have tried to merge all paragraphs related in different sections (lines 35-58; 236-244).
The English language and grammatical errors must be revised throughout the manuscript.
Article has been revised by a professional editing service.
Reviewer 2 Report
This is a study on dietary intervention for the treatment of dry eye. It is well written. Lifestyle & diet are important for dry eye patients so the results are timely.
The following references are applicable to your paper and should be cited or considered for citation.
1) Kawashima, M., et al. Calorie restriction: A new therapeutic intervention for age-related dry eye disease in rats.
https://www.ncbi.nlm.nih.gov/pubmed/20537981?dopt=Citation
2) Kawashima, M, et al. Decreased tear volume in patients with metabolic syndrome: the Osaka study.
https://www.ncbi.nlm.nih.gov/pubmed/24344231?dopt=Citation
3) Kawashima, M, et al. The Association between Dry Eye Disease and Physical Activity as well as Sedentary Behavior: Results from the Osaka Study
https://www.ncbi.nlm.nih.gov/pubmed/25485144
4) Kawashima, M, et al.
Impact of lifestyle intervention on dry eye disease in office workers: a randomized controlled trial.
https://www.ncbi.nlm.nih.gov/pubmed/29618677
Author Response
This is a study on dietary intervention for the treatment of dry eye. It is well written. Lifestyle & diet are important for dry eye patients so the results are timely.
Thank you for your comment.
The following references are applicable to your paper and should be cited or considered for citation.
Thank you for your suggestions. We have taken your concern in consideration and introduction section has been rewriten. The references had been added as detailed below:
[8] Kawashima, M, et al. Decreased tears volume in patients with metabolic syndrome: the Osaka (Lines 47-48)
[10] Kawashima, M, et al. Impact of lifestyle intervention on dry eye disease in office workers: a randomized controlled trial. (Lines 52-54)
[11] Kawashima, M, et al. The Association between Dry Eye Disease and Physical Activity as well as Sedentary Behavior: Results from the Osaka Study. (Lines 52-54)
[12] Kawashima, M., et al. Calorie restriction: A new therapeutic intervention for age-related dry eye disease in rats. (Lines 52-54)
We would like to inform that our manuscript has been revised by a English professional editing service.